# Factors Affecting Zero-Waste Behaviours of College Students

**DOI:** 10.3390/ijerph19159697

**Published:** 2022-08-06

**Authors:** Eun-Hi Choi, Hyunjin Lee, Mi-Jung Kang, Inwoo Nam, Hui-Kyeong Moon, Ji-Won Sung, Jae-Yun Eu, Hae-Bin Lee

**Affiliations:** College of Nursing, Eulji University, Uijeongbu-si 11759, Gyeonggi-do, Korea

**Keywords:** zero waste, health, microplastics, college students

## Abstract

This study evaluated the recognition and attitude toward microplastic and zero waste among college students and investigated the factors influencing their zero-waste behaviours. The study was conducted from 20 August 2021 to 10 September 2021, including students at a university in G metropolitan city, Republic of Korea. A total of 196 data were analysed. Statements were developed to verify how the use of disposables and the recognition, attitude, and behaviours related to zero waste were affected during the COVID-19 pandemic. Family type and usage of disposables were the factors affecting zero-waste behaviour in Model 1. In Model 2, which included the subcategory of zero-waste recognition, the health effects of microplastics and environmental preservation were significant factors. In Model 3, which included the subcategory of zero-waste attitude, the health effects of microplastics (β = 0.149, *p* = 0.016), use of eco-friendly products (β = 0.342, *p* < 0.001), and environmental preservation (β = 0.317, *p* < 0.001) were significant factors. The use of plastic products increased dramatically during the COVID-19 pandemic. Research and education are needed to promote zero-waste behaviours with a focus on microplastics. Raising awareness of the health effects of microplastics can enhance the effectiveness of education.

## 1. Introduction

During the COVID-19 pandemic [1], the World Health Organization (WHO) recommended wearing masks, maintaining social distance, and avoiding gatherings [2]. To maintain social distancing, non-face-to-face classes using online platforms were started, and students attended classes from homes or dorm rooms without attending colleges. Consequently, they ended up spending more time in their living spaces [3] and frequently used home meal replacement or delivery food [4].

Given these changes in lifestyle during the pandemic, the demand for plastic products increased, producing a severe impact on the environment. Plastic waste management was already considered a major environmental issue before the COVID-19 pandemic [5,6]. After the outbreak, the demand for plastic disposables, such as convenience and delivery food containers, has dramatically increased [7], leading serious environmental issues. In particular, the annual usage of plastic per person in Korea is 132.7 kg, which is the highest in the world [8]. However, the current waste management system is not sufficiently effective to manage existing plastic waste [9], and the rapid increase in the amount of plastic waste due to COVID-19 is expected to lead to a bigger problem.

Plastic waste contains many harmful substances, with microplastic being the most hazardous material [10]. In previous studies, microplastics were detected in marine organisms, and harmful effects of microplastics in various aquatic organisms have been reported [11,12]. In addition, microplastics were detected in table salt [13], drinking water [14], and air [15], indicating that human exposure to microplastics is inevitable. Recent studies have reported the association of microplastics with the development of various diseases, including cancer [16,17,18].

These concerns regarding plastic waste led to the creation of a new concept of zero waste. Its definition differs depending on the purpose of the activity and position of the activity subject [19,20,21]. In 2018, the Zero Waste International Alliance (ZWIA) was defined it as ‘the conservation of all resources through responsible production, consumption, reuse, and recovery of all products, packaging, and materials, without burning them and without discharge to land, water, or air, which may threaten the environment or human health’ [19]. According to Hannon and Zaman [21], zero waste is a catalyst that can encourage the participation of local communities to build sustainable cities for the future. Zero waste is considered a concept that goes beyond ‘generating no waste’ and is part of the resource recirculation society, which believes that waste is a resource.

Previous overseas studies related to zero waste were mainly related to industrial field and resource recirculation [21], recycling insurance and pre-recycling methods [22], and zero-waste cities [20,23,24,25], with a focus on building resource recirculation cities with cooperation between governments and industries. However, local research has been limited to passion industries, design, and resource utilisation [26,27,28,29], and there has been no study related to health.

To this end, the present study aims to determine how the recognition and attitude of zero waste affected the behaviour of students who spent relatively more time in their living spaces than other age groups because of online classes during the COVID-19 pandemic. In addition, by examining the current status of microplastic use by college students, it is intended to understand the change in plastic use caused by the pandemic. The purpose of this study was to lay the foundation for the development of programs that promote zero-waste behaviours.

## 2. Materials and Methods

### 2.1. Study Design

This was a cross-sectional study designed to identify how the recognition and attitude of zero waste affected the behaviour of students during the COVID-19 pandemic, when the usage of disposables was increased.

Previous studies have shown that there is a close correlation between knowledge, attitude, and behaviour of a particular object [30]. Knowledge is awareness or understanding learned through education or experience [31]. The more reasonable the content is, the greater the effect on attitudes and behaviours; on the contrary, it is difficult to show consistent attitudes or behaviours if knowledge is insufficient [32]. An attitude is a response that has positive or negative characteristics about a particular object [33]. This attitude affects behaviour [34], but the attitude and the actual behaviour are not always consistent [35].

The model of this study is that the knowledge and attitude of the use of disposable products among Korean college students on the environment affect zero-waste behaviour. Knowledge of zero waste has not been revealed as an experimental study of the human body, and is a result based on changes in marine life or the environment. Therefore, it was more appropriate to use it as recognition rather than the expression of knowledge. The model of this study is shown in Figure 1.

### 2.2. Subjects and Data Collection

The study was conducted from 20 August 2021 to 10 September 2021, including university students in G metropolitan city, Republic of Korea. The convenience sampling method was used to assess subjects who agreed to the study purpose. The questionnaire was formed to answer online and was delivered to college students through SNS. The minimum sample size required for regression analysis was 173 subjects using the G Power 3.1 Programme and considering the significant level of 0.5, power of 0.95, and total predictive factor of 14 in linear multiple regression analysis. Considering the dropout rate as 20%, the survey targeted 207 people, but 197 people actually responded. The survey was completed only when all the online responses were answered, and there were no missing values in the survey response. However, excluding one respondent whose outlier was 3 or higher in the regression analysis, the final analysis was 196 people.

Data were collected following the approval of the Institutional Bioethics Committee of E University (EU21-061). Online surveys were conducted after the study participants were informed of the purpose of the study, and they provided consent for data collection.

### 2.3. Study Tools

#### 2.3.1. General Characteristics

As general characteristics of the study subjects, age, sex, school year, major, and family type were assessed. The majors were categorised as health and medicine, natural science and engineering, education, humanity and social science, and others. The family types were classified as single-member households and two or more-member households.

#### 2.3.2. Change in the Usage of Disposables after COVID-19 Outbreak

To verify the change in the usage of disposables after COVID-19 outbreak, there are few existing studies, so news and articles were searched. Accordingly, the following three questions were asked: ‘After COVID-19 outbreak, do you experience a change in the usage of disposable packing containers?’, ‘After COVID-19 outbreak, do you experience a change in the usage of delivery apps?’ and ‘After COVID-19 outbreak, do you experience a change in usage of parcel delivery service?’. The answers to the questions were ‘increased’ and ‘no change’.

#### 2.3.3. Zero-Waste Recognition Focusing on Microplastics

Statements related to the recognition, attitude and behaviour of zero waste were developed by reviewing the literature [5,6,10,11,12,13,14,15,16,17,18] and searching social networking sites (SNS), the internet [8,9,19], and newspaper articles. The suitability of statements was verified by three professionals and five zero-waste executors. 

Recognition was divided into three categories: the generation process of microplastics, the health impact of microplastics, and environmental preservation. There were five statements regarding the generation process of microplastics: ‘I have never heard of microplastics’, ‘microplastics are generated during the disposal of plastic containers, there are microplastics in toothpaste and cosmetics, microplastics are reproduced by sunlight’, and ‘waste has come a full circle and come to my table’. There were six statements regarding the health impact of microplastics: ‘plastic itself contains carcinogens’, ‘microplastics lead to the accumulation of residual contaminants in the human body’, ‘microplastics cause systemic inflammation and immunosuppression’, ‘intake of microplastics causes cough, laboured respiration, and pulmonary function insufficiency’, ‘microplastics can travel through blood vessels’, and ‘disposable cups contain substances causing an inflammatory response and adenocarcinoma’. There were four statements regarding environmental preservation: ‘I know what zero-waste campaign is, I know environment-friendly enterprises, I know what is a recycle symbol’, and ‘I know some environmental policies, such as collecting empty bottles and tumbler discounts’. Three answers were provided: ‘Yes, No, or Not sure’. ‘Yes’ was assigned one point, and ‘No’ and ‘Not sure’ were assigned zero points. Points for each category were summed. 

#### 2.3.4. Attitude toward Eco-Friendly Products

The attitude was classified into five categories: eco-friendly production by companies, purchasing eco-friendly products, using eco-friendly products, separating disposables, and environmental campaigning. There were two statements regarding eco-friendly production by companies: ‘It is important to make products from materials that can be recycled’ and ‘a company should make eco-friendly products’. There were four statements on purchasing eco-friendly products: ‘I think the things that I do not need are trash, I think the more eco-friendly products are, the better, carrying something such as a tumbler is inconvenient, and it is important to use less disposable packaging’.

There were three statements on using eco-friendly products: ‘It is boring to use purchased products for a long time’, ‘It is convenient to use straws, wooden chopsticks, and plastic bags’, and ‘It is convenient to use disposable wet wipes’. There were five statements on separating disposables: ‘eventually, it is beneficial for me to reduce the usage of disposable containers’, ‘the problem of disposable waste does not directly affect me’, ‘it is meaningless to make an effort to reduce the usage of disposables’, and ‘I feel uncomfortable generating plastic waste’. There were two statements on environmental campaigning: ‘I closely follow environmental campaigns’ and ‘I have thought about participating in an environmental campaign’.

Each statement was answered based on a 5-point Likert scale, with ‘strongly disagree’ assigned one point and ‘strongly agree’ assigned five points. Among the statements, negative responses for attitude toward eco-friendly products were processed as reverse statements. Average scores for each category are presented. A higher score indicated a positive attitude. Cronbach’s alpha for this tool was 0.731.

#### 2.3.5. Zero-Waste Behaviours

Behaviour was classified into four categories: purchasing eco-friendly products, using eco-friendly products, separating disposables, and environmental campaigns. There were five statements on purchasing eco-friendly products: ‘I check the recycle mark before buying something’, ‘I reduce waste by only purchasing what I need’, ‘I purchase products using as less disposable packaging as possible’, ‘I use eco-friendly products if it is possible’, and ‘if possible, I select no disposable check box when I order delivery food’. There were four statements on using eco-friendly products: ‘I keep using a product once I purchase it, I reuse daily necessities with containers by refilling them, I try not to use disposable wet wipes’, and ‘I do not use disposables when I have reusable dishware’. There were four statements on separating and sending out disposables: ‘I actively separate and send out food and plastics, I try not to use delivery apps and parcel delivery services as much as possible because they generate much disposable waste, I remove the plastic packaging of PET bottles before taking them out to prevent generating mixed waste’, and ‘I empty and clean recyclable plastic items before taking them out’. There were two statements on environmental campaigns: ‘I participate in empty bottle collection and tumbler discount’ and ‘I reduce disposable waste by using reusable shopping bags’.

Each statement was answered based on a 5-point Likert scale. Cronbach’s alpha for this tool was 0.767.

### 2.4. Data Analysis

Statistical analyses were conducted using SPSS (version 26.0; IBM SPSS Statistics, New York, NY, USA). The general characteristics and variables of the participants are presented as means and standard deviations or frequencies and percentages. The subjects’ general characteristics and differences in zero-waste behaviour depending on the usage change in disposables during COVID-19 were analysed using a t-test and analysis of variance (ANOVA), respectively, followed by Scheffé’s post-hoc test. Hierarchical multiple regression analysis was used to investigate the effect of zero-waste behaviour, controlling for sex, age, school year, and major. In Model 1, the family type and usage change in disposables after the COVID-19 outbreak were entered. In Model 2, zero-waste recognition was entered as a subcategory in Model 1. In Model 3, zero-waste attitude was entered as a subcategory in Model 2.

## 3. Results

### 3.1. General Characteristics of the Study Subjects

A total of 196 participants, including 34 men (17.3%) and 162 women (82.7%), were included. The mean age of participants was 20.9 years of age. Regarding the school year, there were 48 (24.5%) first-year students, 33 (16.8%) second-year students, 79 (40.3%) third-year students, and 36 (18.3%) fourth- or higher-year students. Regarding major, 45 (35.2%) participants studied health and medicine, 36 (18.4%) studied natural science and engineering, 45 (23%) studied education, and 82 (39.3%) studied anthropology, sociology, and arts. Regarding family type, 25 participants (12.8%) lived alone and 171 (87.2%) lived with their families. Regarding the usage of disposables due to COVID-19, 136 (69.4%) participants reported increased usage. Regarding the usage of delivery apps due to COVID-19, 135 (68.9%) participants reported increased usage. Regarding the usage of parcel delivery services, 134 (68.4%) participants reported increased usage (Table 1).

### 3.2. Zero-Waste Recognition, Attitude, and Behaviour

The recognition score was provided as 0 or 1 for each statement. Each Mean ± SD for sub-category of recognition was the Mean ± SD of the sum of the contained item scores.

Regarding zero-waste recognition, the score for the microplastics generation process was 3.1/5 points and that for environmental preservation was 2.6/4 points. The score for the health effects of microplastics was 3.1/6. The average score for this category was 8.8/15.

Regarding attitude toward eco-friendly products, the score for eco-friendly production by companies was 4.4/5 points, that for purchasing eco-friendly products was 3.7/5 points, that for using eco-friendly products was 2.8/5 points, that for separating disposables was 4.1/5 points, and that for environmental campaigning was 3.6/5 points. The average score for each category was 3.7/5.

Regarding zero-waste behaviour, the score for purchasing eco-friendly products was 3.5/5 points, that for using eco-friendly products was 3.7/5 points, that for separating disposables scored 4.0/5 points, and that for environmental campaigning was 3.6/5 points. The average score for the category was 3.7/5 points (Table 2).

### 3.3. General Characteristics and Differences in Zero-Waste Behaviour Due to Usage Change in Disposables during COVID-19 

Zero-waste behaviours, depending on the general characteristics of the study subjects, revealed significant differences by sex (t = −3.632, *p* = 0.001) and family type (t = −2.324, *p* = 0.021). Zero-waste behaviours showed significant differences in the usage of disposables during COVID-19 (t = −2.454, *p* = 0.015) (Table 1).

### 3.4. Factors Influencing Zero-Waste Behaviour

Hierarchical multiple regression analysis was conducted to identify the factors affecting zero-waste behaviour (Table 3). Sex, age, school year, and major were entered in each hierarchical regression model. The Durbin–Watson statistic was used to assess multicollinearity for verifying the basic assumption of regression analysis. The Durbin–Watson value was 2.232, and multicollinearity was low (variance inflation factor (VIF) < 5).

In Model 1, family type (β = 0.146, *p* = 0.042) and disposable usage (β = 0.158, *p* = 0.049) were significant factors. The regression model was significant (F = 3.540, *p* < 0.001) and the explanatory power was 11.5%.

Model 2 included the subcategories of zero-waste recognition. The health effects of microplastics (β = 0.197, *p* = 0.008) and environmental campaign (β = 0.236, *p* = 0.001) were significant factors. The regression model was significant (F = 5.185, *p* < 0.001), and the explanation power was 21.8%.

In Model 3, the subcategories of zero-waste attitude were included. The health effects of microplastics (β = 0.149, *p* = 0.016), use of eco-friendly products (β = 0.342, *p* < 0.001), and environmental campaign (β = 0.317, *p* < 0.001) were significant factors. The regression model was significant (F = 10.730, *p* < 0.001), and the explanation power was 47.3%.

## 4. Discussion

The present study was undertaken to lay a foundation for programme development aimed at improving zero-waste behaviour by identifying the effects of zero-waste recognition and attitude. This study aimed to discuss focusing on the sub-factors that influence behaviour among zero-waste recognition and attitude.

In our analysis, the score for the health effects of microplastics was the lowest, at 3.1/6 points, in the zero-waste recognition category. Moreover, only the recognition of health effects of microplastics affected zero-waste behaviour amongst the recognition categories. In other words, recognition of the adverse link between microplastics and health was a major factor promoting zero-waste awareness and behaviour. Various adverse effects of microplastics on marine organisms have been reported [11,12]. Contamination of natural resources by microplastics enables their entry into the food chain and thus into the human body [29]. The accumulation of microplastics in human liver, kidney, and intestines disrupts energy and lipid metabolism [36]. However, the participants of the present study showed a low level of recognition regarding the health effects of microplastics. Thus, health information related to microplastics should be provided. However, such education should be based on empirical studies of microplastics and its health risk. Animal experiments and studies have proven the risk of microplastics in other organisms [37,38]; however, no study has confirmed the health hazards in humans. This is because studies on the environment and its health effects are extensive, and the health risk appears over the long term rather than the short term. Furthermore, health risk differs amongst individuals, rendering the identification of health problems caused by the environment difficult [39]. Studies collecting fundamental data for building scientific recognition in the long term are warranted to promote awareness regarding the health risks of microplastics.

In the category of zero-waste attitude, eco-friendly production of companies achieved the highest scores, whilst the usage of eco-friendly products achieved the lowest scores. The usage of eco-friendly products includes using the purchased products for a long time, convenience of using disposables, and convenience of cleaning with disposable wet wipes. According to a study on environmental problems by IPSOS [40], 91% of Korean respondents answered that there was concern regarding packaging waste and using disposables that cause environmental pollution. On the other hand, in terms of individual behaviour to reduce unrecyclable packing materials, 27% of respondents answered that they tried to minimise their use by overcoming the habit of buying disposables, which was a low percentage, consistent with the results of the present study. In other words, people know that the use of disposables should be reduced, but they find it difficult to use recycled products. In another study, respondents answered that they occasionally use disposables because they are convenient and cheap. Nonetheless, with increased awareness of the environmental regulation policy, respondents tried avoiding the use of disposables to protect the environment [41]. In 2018, the Ministry of Environment announced a plan aimed at shifting the linear economic structure involving production and disposal to a circulating system involving production and recycling by 2027 [42]. Accordingly, regulatory measures were enacted to restrict the use of replaceable disposables and minimise unnecessary excessive packaging. However, this policy was modified and the use if disposables was allowed during the COVID-19 pandemic [42]. Accordingly, the usage of disposables increased due to COVID-19, and compliance with zero-waste behaviours became challenging as the non-use of disposables led to customer inconvenience. Therefore, more promotions and campaigns are required to encourage people to change their habits and inculcate zero-waste behaviours. Furthermore, companies should develop various alternative products that customers can use conveniently, such as tumblers, and offer them with a wide range of choices. 

The present study showed that more interest in environmental campaigns and a positive attitude towards participating in such campaigns led to positive effects on zero-waste behaviours. Environmental campaigns and education are linked to students’ eco-friendly attitudes [43,44]. Recently, Korea introduced environmental issues to the educational curriculum [45]. In elementary schools, education on environmental pollution is mandatory, although there is insufficient recognition of environmental practices and participation in middle and high school curricula [46]. Furthermore, in the Korean education system, which focuses on college entrance examinations, it is difficult for students to acquire information on environmental issues and to have the opportunity to think about them by themselves. Therefore, students must be offered more opportunities to gain sufficient recognition about environmental issues. This will help them acquire reliable information, promote zero-waste behaviours, and foster thinking on minimising environmental pollution or microplastic usage. Additionally, various exciting public relations campaigns should be developed to encourage people.

According to a previous study, the increased usage of delivery food due to COVID-19 has altered eating habits [3]. Students had to use delivery apps and disposable containers to avoid contracting COVID-19; this increase is expected to be temporary during the COVID-19 pandemic. A negative impact was expected on zero-waste behaviour owing to increasing disposable plastic use and shipping apps as a result of COVID-19. However, based on the results, the increased use of disposables, delivery apps, and parcel delivery services as a result of COVID-19 did not affect zero-waste behaviour. However, according to the Institute of Medicine, more infectious diseases are expected to emerge, indicating other possible outbreaks in the future [47]. Given the prolonged COVID-19 pandemic, awareness regarding the health effects of microplastics, importance of using recyclable products, and concerns for the environment should be promoted.

There were some limitations in the present study: First, the study subjects were limited to a particular age group of college students rather than all age groups. College students are the most intelligent, and other factors may be added for different age groups. As the subjects in this study were primarily women, it was included as a control factor in the regression analysis; however, this was also a limitation of the study. Second, although the measurement tool was developed with much effort, but it could not assess the comprehensive effect of microplastics and waste. Therefore, additional tools should be created. Finally, this was a cross-sectional study undertaken at a specific time when the COVID-19 pandemic extended. Additional studies on zero-waste behaviours in the long term are warranted. Despite these limitations, the increased use of plastic products during the COVID-19 pandemic is thought to have an impact on the environment. In this study, raising recognition about the effects of microplastics on health will reduce plastics and contribute to increasing the efficiency of education that can promote zero-waste behaviour. Zero-waste behaviours may vary depending on age and presence/absence of chronic diseases. Future studies should include other subject groups. In recent years, the infertility rate has been rising, while the birth rate has reduced. Therefore, additional studies are required to verify the link between reproductive health and microplastics in the environment.

## 5. Conclusions

Based on the results of previous studies that microplastics have a negative effect on health, this study identified factors affecting the zero-waste behaviour of college students. Zero-waste recognition focusing on microplastics, attitudes toward eco-friendly products and zero-waste behaviour were evaluated, and the relationship between these variables was verified. Zero-waste behaviour was found to be related to recognition of the health impact of microplastics and attitudes toward separating disposables. In order to improve the zero-waste behaviour of college students, it means that an education program is needed to improve the awareness of microplastic health effects and attitudes towards separating disposables.

It is expected that the results of this study will perform as a basis for the development of research and education programs to improve zero-waste behaviours of college students in the future, and we suggest the following: First, to encourage zero-waste behaviour, college students need to recognize the importance of microplastics’ health effects and attitudes toward the segregation of disposables. Therefore, it is necessary to provide an education program that can improve the health risks of microplastics and the attitude toward the separation of disposable products. Second, it can be effective to improve the awareness and attitude of university students through collaboration with universities. It is proposed to build and operate a system that provides zero-waste-related education programs and campaigns with universities. Third, we propose to develop a zero-waste program including the health effects of microplastics and the importance of separating disposables and to verify the effects. Fourth, a campaign strategy tailored to all age groups, from infants to the elderly, can contribute to the spread of zero-waste behaviours. However, this study has a limitation in that it did not expand the investigation of factors affecting the zero-west behaviours of various age groups. We propose to study the factors influencing zero-waste behaviour across different age groups.

## Figures and Tables

**Figure 1 ijerph-19-09697-f001:**
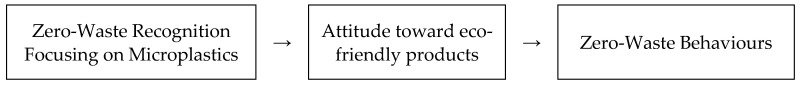
A model of this study.

**Table 1 ijerph-19-09697-t001:** General characteristics and differences in zero-waste behaviours depending on the usage change in disposables during COVID-19 (n = 196).

Variable	Category (N/%)	Mean ± SD	t/F	*p*
Sex	Male (34/17.3)	3.1 ± 0.5	−3.632	0.001
	Female (162/82.7)	3.7 ± 0.4		
Age (years)	18 (18/9.2)	3.5 ± 0.4	1.391	0.220
	19 (29/14.8)	3.6 ± 0.5		
	20 (34/17.3)	3.8 ± 0.4		
	21 (46/23.5)	3.8 ± 0.4		
	22 (30/15.3)	3.6 ± 0.4		
	23 (22/11.2)	3.6 ± 0.6		
	≥24 (17/8.7)	3.8 ± 0.4		
School year	First year (48/24.5)	3.6 ± 0.4	2.115	0.100
	Second year (33/16.8)	3.7 ± 0.5		
	Third year (79/40.3)	3.7 ± 0.4		
	Fourth year and above (36/18.4)	3.8 ± 0.4		
Major	Health and medicine (33/16.8)	3.8 ± 0.5	1.384	0.249
	Natural science and engineering (36/18.4)	3.7 ± 0.4		
	Education (45/23.0)	3.6 ± 0.5		
	Humanities, social sciences, and arts (82/39.3)	3.7 ± 0.4		
Family type	Living alone (25/12.8)	3.9 ± 0.5	−2.324	0.021
	Living with family (171/87.2)	3.7 ± 0.4		
Usage of disposable packing containers	No change (60/30.6)	3.6 ± 3.6	−2.454	0.015
	Increase (136/69.4)	3.7 ± 0.4		
Usage of delivery apps	No change (61/31.1)	3.7 ± 0.4	−0.583	0.560
	Increase (135/68.9)	3.7 ± 0.5		
Usage of parcel delivery services	No change (62/31.6)	3.7 ± 0.4	−0.499	0.619
	Increase (134/68.4)	3.7 ± 0.5		

**Table 2 ijerph-19-09697-t002:** Zero-waste recognition, attitude, and behaviour (n = 196).

Variable		Mean ± SD	Range
Recognition	Microplastic generation process	3.1 ± 1.2 *	0–5
	Health effects of microplastics	3.1 ± 0.8 *	0–6
	Environment protection	2.6 ± 0.8 *	0–4
	Total recognition	8.8 ± 3.1 *	0–15
Attitude	Eco-friendly production of companies	4.4 ± 0.6	1–5
	Purchasing eco-friendly products	3.7 ± 0.5	1–5
	Using eco-friendly products	2.8 ± 0.6	1–5
	Separating disposables	4.1 ± 0.6	1–5
	Environmental campaigns	3.6 ± 0.8	1–5
	Total attitude	3.7 ± 0.4	1–5
Behaviour	Purchasing eco-friendly products	3.5 ± 0.6	1–5
	Using eco-friendly products	3.7 ± 0.5	1–5
	Separating disposables	4.0 ± 0.6	1–5
	Environmental campaigns	3.6 ± 0.9	1–5
	Total behaviour	3.7 ± 0.5	1–5

* Each Mean ± SD for recognition variables was the Mean ± SD of the sum of each item score.

**Table 3 ijerph-19-09697-t003:** Factors influencing zero-waste behaviour.

Variable		Model 1				Model 2				Model 3			
		β	t	*p*	VIF	β	t	*p*	VIF	β	t	*p*	VIF
Family type (ref = living with family)		0.146	0.899	0.042	1.113	0.116	1.721	0.087	1.125	0.021	0.361	0.718	1.196
Usage of disposable containers (ref = no change)		0.158	0.714	0.049	1.400	0.123	1.619	0.107	1.436	0.073	1.134	0.259	1.528
Usage of delivery apps (ref = no change)		−0.096	0.664	0.247	1.506	−0.100	−1.281	0.202	1.521	−0.009	−0.138	0.890	1.577
Usage of parcel delivery (ref = not change)		0.047	0.863	0.517	1.159	0.048	0.704	0.483	1.161	0.012	0.217	0.829	1.192
Recognition	Microplastic generation process					0.044	0.615	0.540	1.306	−0.024	−0.407	0.685	1.339
	Health effects of microplastics					0.197	2.667	0.008	1.365	0.149	2.443	0.016	1.384
	Environmental campaign					0.236	3.300	0.001	1.272	0.098	1.523	0.130	1.531
Attitude	Eco-friendly production of companies									0.036	0.548	0.584	1.594
	Purchasing eco-friendly products									0.053	0.858	0.392	1.409
	Using eco-friendly products									0.342	6.122	<0.001	1.153
	Separating disposables									0.099	1.530	0.128	1.549
	Environmental campaign									0.317	4.745	<0.001	1.654
R2		0.161				0.270				0.522			
Adjusted R2		0.115				0.218				0.473			
F		3.540				5.185				10.730			
*p*		<0.001				<0.001				<0.001			
Durbin–Watson statistic										2.232			

## Data Availability

All relevant data are within the paper.

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
