# Peer review of "Factors Affecting Zero-Waste Behaviours of College Students"

_ijerph, 2022, doi:10.3390/ijerph19159697_

Round 1

Reviewer 1 Report

2. Materials and Methods 72

2.1. Subjects and Data Collection

R=74-75 - The study was conducted from August 20, 2021, to September 10, 2021, including university students in G metropolitan city (Who else? and how the questionnaire reaches to respondents?)

R= 78- 81 - Considering 20% dropout rate, the survey was conducted on 197 subjects, except for 1 subjects of the respondents who showed abnormality, and the data of 196 subjects were analysed.

Still couldn’t understand! How many total data was collected in survey study? Because Considering 20% dropout rate but only mentioned that the survey was conducted on 197 subjects and the data of 196 subjects were analysed (it’s means only 1 respondent who showed abnormality, its become 20% dropout rate?)

2. Materials and Methods

2.2. Study Tools  

2.2.1. General Characteristics

2.2.2. Change in the Usage of Disposables After COVID-19 Outbreak

 My previous comment didn’t answer well, the orders of the presentation makeup, (I don’t see clearly the theoretical background of this study and no any reference on it.)

2.2.2. Change in the Usage of Disposables After COVID-19 Outbreak

R 95………two answers were provided, ‘decrease and no change’ and ‘increase’.?

2.2.3. Zero-Waste Recognition Focusing on Microplastics

R= 98- 99 Statements related to the recognition of zero waste focusing on microplastics were developed by reviewing the literature and searching social networking sites (SNS), the Internet, and newspaper articles (where is the reference?) and also others reference for related study and other part following of study design.

l  Suggestion theoretical and study design

Study design – I suggest that just make one table and measurements tool not rewrite for example: Each statement was answered based on a 5-point Likert scale, with ‘strongly disagree’ assigned one point and ‘strongly agree’ assigned five points.

Theoretical – explain clearly your study and related references

3. Results

3.2. Zero-Waste Recognition, Attitude, and Behaviour

Table 1. not match method design

2.3. Data Analysis

Statistical analyses were conducted using SPSS (version 26.0; IBM SPSS Statistics, 171 NYC, USA). The general characteristics and variables of the participants are presented as means and standard deviations or frequencies and percentages.

Note: if the method is well design and data analysis the results will no problem in all table conducted using SPSS (version 26.0; IBM SPSS Statistics, NYC, USA) I see still not very confident significantly for example: 2.2.3. Zero-Waste Recognition Focusing on Microplastics (Three answers were provided: ‘Yes, No, or Not sure’. ‘Yes’ was assigned one point, and ‘No’ and ‘Not sure’ were assigned zero points. Points for each category were summed.) where is the results? In Table 1 recognition didn’t match design (Three answers were provided: ‘Yes, No, or Not sure’. ‘Yes’ was assigned one point, and ‘No’ and ‘Not sure’ were assigned zero points)

4. Discussion and 5. Conclusions

This also will affect by the study method (survey design) and the results)

All above comments need more time for Autor to modify I suggests the author (focus on study design part of study methodology, Theoretical of yours and others related references and data analysis)

2022/7/3

Author Response

Thank you for your helpful comments on this manuscript. We tried to address the issues that you pointed out in your comments as much as possible, and the revised parts of the manuscript are shown in blue font in the Word file. 

Reviewer 2 Report

I find the revisions done somewhat superficially. For instance, Conclusions are still very brief. I think that Discussion and Conclusions could better talk with the results. What sorts of conclusions can one really make based on your research? For instance, what sort of "programme development" can be done based on your results?

- Abstract: This sentence could come later in Abstract: "Our results lay the foundation for programme development aimed at promoting zero-waste activities."

- I still cannot find "Republic of Korea" mentioned in Abstract nor in section 2.1 (see, your response to reviewer).

- What is meant with "abnormality" on line 80? I would still give better reasons for why the answers of this one student were ignored.

- "By the way" on line 341 is not an academic expression.

Author Response

(The authors gave the same response as above.)

Reviewer 3 Report

The topic is very interesting and very important. In my opinion, the scientific value should be slightly increased. Be clear about the hypotheses. Have them verified. Relate research to the theory of economics, especially behavior.

Author Response

(The authors gave the same response as above.)

Round 2

Reviewer 1 Report

Comments and suggestions to author

1.     Subjects and Data Collection

Page 3 R126-127 (To make sure the sentence you write on (Statements related to zero- waste attitudes and behaviour were also developed in the same way) please recheck of the below otherwise the sentence is confusing

2.3.3. Zero-Waste Recognition Focusing on Microplastics

R 142-144 (Three answers were provided: ‘Yes, No, or Not sure’. ‘Yes’ was assigned one point, and ‘No’ and ‘Not sure’ were assigned zero points. Points for each category were summed.

2.3.4. Attitude toward eco-friendly products

R 167-168 (Among the statements, negative responses for zero waste were processed as reverse statements. Average scores for each category are presented. A higher score indicated a positive attitude).

Note: look the tittle on this sentence and measurements development.

2.     Results in the Table 2

Recognition – the measurements design on 2.3.3. Zero-Waste Recognition Focusing on Microplastics (Three answers were provided: ‘Yes, No, or Not sure’. ‘Yes’ was assigned one point, and ‘No’ and ‘Not sure’ were assigned zero points. Points for each category were summed. But how could be the results study on Table 2 in Recognition Microplastic generation process 3.1 ± 1.2* 0–5

Health effects of microplastics 3.1 ± 0.8* 0–6

Environment protection 2.6 ± 0.8* 0–4

Total recognition 8.8 ± 3.1* 0–15

Doesn’t make sense with study measurements design

So, this related to my comments on number 1 above.

Note: Please check again and revise the measurement design that can match the results of the study.

8/3/2022

Author Response

Thanks for the reviewer's comments. Thank you for your attention to detail, so we had a chance to learn a lot. 

This manuscript is a resubmission of an earlier submission. The following is a list of the peer review reports and author responses from that submission.

Round 1

Reviewer 1 Report

 Comments and Suggestions to Author

Abstract

  1. Didn’t see clear what the purpose of study
  2. R 15-20 should be writing the numbers?

  1. Introduction

The study purpose didn’t strong. For example in the R61-62 sentence state that,…

  1. Materials and Methods

2.1. Study Design

Study Design didn’t enough information

2.2. Subjects and Data Collection

1.Data collections didn’t enough information and makeup, which cause didn’t clear for the reader

  1. R81-82, How the Author get this number (Considering 20% dropout rate, the survey was conducted on 197 subjects, and the data of 196 subjects were analysed.)

2.3. Study Tools

Makeup with the theoretical

Materials and Methods - Suggestion this part still need to reorder to make study clear and not makeup.

The results

Didn’t match the study design – the results didn’t enough information to support study.

For example, we see just Table 4. Factors influencing zero-waste behavior

  1. Discussion, 5. Conclusions, need modified that can match the study design and result.
  2. Study contribution no clear?
  3. Future and limitations study where?

I concluded that the study needs more time to modifications which can match the rigid and significance of the publications.

Have good luck

2022/5/15

Reviewer 2 Report

Main title of the manuscript is sharp and well formulated.

Introduction is also well written and consistent describing the changes in students' lifestyles due to COVID19 pandemic and giving attention to building sustainable cities for the future.

In Abstract and in sections 2.4 and 3.3, an expression "after the COVID-19 pandemic" is used, but is the pandemic over yet? Should it be "during" instead?

In Abstract, "a university in G metropolitan city" is mentioned. The authors could add the context of the study to Abstract and to section 2.1; that is, the study was executed namely in Republic of Korea. Since section 2.1 is very short, the authors could present the main research question in section 2.1. or elaborate more on the study aim in section 2.1.

In section 2.2 and in Abstract it is stated that "the survey was conducted on 197 subjects, and the data of 196 subjects were analysed". Please, give reasons in the article why the answers of one of the students are not included in the research.

For instance in section 2.3.3, the authors present "questions" regarding the themes of the research. Are these really "questions" that were presented to the participants or could they be called "statements" or "claims"? In addition, consider presenting the questions/statements/claims in sections 2.3.2, 2.3.3, 2.3.4 and 2.3.5 in Tables and not as part of the body text.

I wonder if Cronbach's alphas (2.3.4 and 2.3.5) should be mentioned in Results and not in Materials and Methods.

There is no need for verbalizing section 3.1 because the same information can be found in Table 1. In other words, there is no need to repeat the same information twice: both in written text (as part of the body text) and in a Table (applies to other sections, as well). The Tables are sufficient.

I wonder if Table 1 presents results in the first place (apart from the last three items); that is, I would suggest moving Table 1 to section 2.2 in which the participants are presented. The results of the last three items in Table 1 (i.e., usage of disposable packing containers due to COVID-19, usage of delivery apps due to COVID-19, and usage of parcel delivery services due to COVID-19) can be left as they are in Table 1. However, I wonder in which section is the question/statement/claim of "usage of disposable packing containers due to COVID-19" presented to the readers? The other two statements are presented in section 2.3.2.

In Table 2, there is a misspelling regarding "Environment protection m)".

In Discussion (on page 9), it is stated that "the present study showed that more interest in environmental campaigns and a positive attitude towards participating in such campaigns led to positive effects on zero-waste behaviour". The writers could substantiate this argument in more depth because there are also research showing that education for sustainability does not necessary increase people's environmentally responsible behavior.

Conclusions could be more thorough. The authors could, for instance, elaborate more on the manufacturing processes of plastic items and their impact on climate change. Moreover, the division of responsibilities between Global North and Global South regarding recycling and reducing the use of disposables could be taken into consideration, as well.

Reviewer 3 Report

The purpose of the research is interesting. Research hypotheses should be carefully defined. The article is rather about pro-ecological behavior. The current title is too narrow. A very small analysis of the literature on the pro-environmental behavior of young people. I encourage you to read:

  • BaliÅ„ska Agata, Jaska Ewa, Werenowska Agnieszka: The Role of Eco-Apps in Encouraging Pro-Environmental Behavior of Young People Studying in Poland, Energies, vol. 14, nr 16, 2021, s. 1-16, DOI:10.3390/en14164946,
  • Parzonko Anna, BaliÅ„ska Agata, Sieczko Anna: Pro-Environmental Behaviors of Generation Z in the Context of the Concept of Homo Socio-Oeconomicus, Energies, vol. 14, nr 6, 2021, s. 1-18, DOI:10.3390/en14061597,

The sample taken for the research is very small. What factors influenced such a sample selection?